# Roles of Human Liver Cytochrome P450 Enzymes in Tenatoprazole Metabolism

**DOI:** 10.3390/pharmaceutics15010023

**Published:** 2022-12-21

**Authors:** Thien-Kim Le, Young Jin Park, Gun Su Cha, Fikri A. R. Hardiyanti Oktavia, Dong Hyun Kim, Chul-Ho Yun

**Affiliations:** 1School of Biological Sciences and Technology, Chonnam National University, 77 Yongbongro, Gwangju 61186, Republic of Korea; 2Department of Pharmacology and Pharmacogenomics Research Center, Inje University, College of Medicine, Bokjiro 75, Busanjin-Gu, Busan 47392, Republic of Korea; 3Namhae Garlic Research Institute, 2465-8 Namhaedaero, Namhae-gun, Gyeongsang-namdo 52430, Republic of Korea; 4School of Biological Sciences and Biotechnology, Graduate School, Chonnam National University, 77 Yongbongro, Gwangju 61186, Republic of Korea

**Keywords:** carbon hydroxylation, CYP2C19, CYP3A4, human drug metabolite, drug metabolic pathway, tenatoprazole, tenatoprazole sulfide

## Abstract

Tenatoprazole, a newly developed proton pump inhibitor candidate, was developed as an acid inhibitor for gastric acid hypersecretion disorders such as gastric ulcer and reflux esophagitis. It is known that tenatoprazole is metabolized to three major metabolites of 5′-hydroxy tenatoprazole, tenatoprazole sulfide, and tenatoprazole sulfone in human liver, primarily catalyzed by CYPs 2C19 and 3A4. While CYP2C19 prefers the hydroxylation of tenatoprazole at C-5′ position, CYP3A4 is mainly involved in sulfoxidation reaction to make tenatoprazole sulfone. Tenatoprazole sulfide is a major human metabolite of tenatoprazole and is formed spontaneously and non-enzymatically from tenatoprazole. However, its metabolic fate in the human liver is not fully known. Furthermore, no systematic metabolic study has been performed to study tenatoprazole or tenatoprazole sulfide. Here, we studied the functions of human cytochromes P450 in the metabolic pathway of tenatoprazole and tenatoprazole sulfide by using recombinant human P450s and human liver microsomes. Both CYP 2C19 and CYP3A4 showed distinct regioselective and stereospecific monooxygenation activities toward tenatoprazole and tenatoprazole sulfide. Furthermore, a new major metabolite of tenatoprazole sulfide was found, 1′-*N*-oxy-5′-hydroxytenatoprzole sulfide, which has never been reported. In conclusion, the metabolic fates of tenatoprazole and tenatoprazole sulfide should be considered in the clinical use of tenatoprazole.

## 1. Introduction

Tenatoprazole is a drug candidate as a proton pump inhibitor (PPI) (Figure 1). It is under clinical test as a potential therapy for reflux oesophagitis and peptic ulcers [1,2,3]. In terms of chemical structure, tenatoprazole contains an imidazopyridine ring on the benzimidazole moiety that is found in other PPIs such as lansoprazole and omeprazole (Figure 1) [4]. Unlike other PPIs, such as lansoprazole and omeprazole, tenatoprazole has a remarkably prolonged continuation of action, resulting from its plasma half-life being around seven (7) times longer than those of other PPI drugs. Thus, the collected clinical data show that tenatoprazole allows degrees of symptom relief and cure of gastric lesions better than those performed by other PPIs of the same therapeutic category. Therefore, tenatoprazole may be usable for treating non-typical and oesophageal symptoms of dyspepsia, digestive bleeding, and gastro-oesophageal reflux [1].

In vitro pharmacokinetic and metabolism studies have shown a significant difference in the metabolism (*V*_max_ values) of the (+)-(*R*) versus the (−)-(*S*) isomers of human CYP2C19 [5]. The *S*-isomer is metabolized around seven times more slowly than the *R*-isomer. Consequently, the *S*-isomer of tenatoprazole has a much longer mean residence time (MRT) in the human body, as compared with the *R*-isomer. The *S*-isomer is primarily oxidized via CYP3A4, which can compensate for a blockade or potential deficiency of CYP2C19. The *R*-isomer is metabolized via two pathways: primarily by CYP2C19 and, to a lesser extent, CYP3A4. The major human metabolites of tenatoprazole include 5′-OH tenatoprazole, tenatoprazole sulfide, and tenatoprazole sulfone [6]. Among P450 enzymes, CYP 2C19 and 3A4 catalyze monooxygenation reactions of tenatoprazole to 5′-OH tenatoprazole and tenatoprazole sulfone, respectively. Although tenatoprazole is metabolized primarily by the CYP2C19 and CYP3A4 [6], its metabolic pathway has not been investigated in detail in human liver microsomes (HLMs). Furthermore, the metabolic fate of tenatoprazole sulfide, a major metabolite that is spontaneously and non-enzymatically formed from tenatoprazole [7], has never been studied before. 

Drugs frequently fail during the development process due to safety problems associated with the metabolites, rather than the drug candidates themselves. Therefore, guidance from the U.S. Food and Drug Administration (FDA) in 2008 (and revised in 2016) and E.U. International Committees on Harmonization (ICH) M3(R2) recommends that all significant major metabolites should be characterized for evaluation of drug efficacy and toxicity when developing drugs [8,9,10]. Note that the currently generally accepted prevalence threshold for a corresponding metabolite is ≥10% of estimated total drug-related exposure at the level of steady-state. Pharmacologic activity of the metabolite is a significant factor for safety issues. Furthermore, as non-active pharmacological metabolites can be related to toxicity, they must also be evaluated in the overall safety [11].

The specific aim of this investigation was to study the roles of human P450s in the metabolic fates of tenatoprazole and its sulfide, a major metabolite of tenatoprazole, by using HLMs and recombinant human P450s. Both CYP 2C19 and 3A4 showed distinct regioselective monooxygenation activities toward tenatoprazole and its sulfide metabolite to produce 5′-OH tenatoprazole, 5′-OH tenatoprazole sulfide, and tenatoprazole sulfone. Surprisingly, a new major metabolite of tenatoprazole sulfide was found: 1′-*N*-oxy-5′-hydroxytenatoprzole sulfide, which has never been reported before.

## 2. Materials and Methods

### 2.1. Materials

Tenatoprazole (as a racemic mixture) was obtained from Abcam Biochemicals. Omeprazole (racemic mixture), oxidized nicotinamide adenine dinucleotide phosphate (NADP^+^), glucose 6-phosphate, and glucose 6-phosphate dehydrogenase were obtained from Sigma-Aldrich (Saint Louis, MO, USA). Tenatoprazole sulfide was purchased from 4Chem Laboratory (Suwon, Gyeonggi-do, Republic of Korea). 5′-OH tenatoprazole and 5′-OH tenatoprazole sulfide were prepared as described previously [7]. The other chemicals used in this study were of the highest purity grade commercially available.

Pooled liver microsomes from human samples were obtained from ThermoFisher Scientific (Waltham, MA, USA). Recombinant human P450s were heterologously expressed in *Escherichia coli* with a pCW vector containing human P450 cDNA (CYP 3A4, 2C9, 2C19, 1A2, 1B1, 2E1, 2D6, and 2A6) and rat NADPH-P450 reductase (CPR) [12,13,14,15,16,17,18]. Membrane fractions expressing both P450 and CPR of *E. coli* were prepared as described previously [12,13] and were used for the catalytic activity assays.

### 2.2. Hydroxylation of Tenatoprazole and Tenatoprazole Sulfide by Human Liver Microsomes

A racemate of tenatoprazole was used in this work because the chiral tenatoprazoles currently are not commercially available. The reactions using HLMs contained incubation at 37 °C in 0.25 mL mixtures containing P450 (0.5 mg protein/mL) in 100 mM potassium phosphate buffer (pH 7.4), an NADPH-generating system (NGS, final concentration of 10 mM glucose-6-phosphate, 0.50 mM NADP^+^, 1.0 IU yeast glucose-6-phosphate dehydrogenase per mL), and 0.1 mM of tenatoprazole or tenatoprazole sulfide. Incubations generally were performed for 2 h in the presence or absence of the NGS. Tenatoprazole and its metabolites were extracted twice using cold ethyl acetate as an organic solvent. Then centrifugation (1000× *g*, 20 min) was done to separate the organic layer from the aqueous layer. Omeprazole was added to this solution as an internal standard (IS) at a final 20 μM concentration [7]. The organic layer was concentrated under nitrogen gas stream. The dried residues were then dissolved in the mobile phase (180 μL) and 30 μL was injected onto the column.

High-performance liquid chromatography (HPLC) was performed on the basis of a previously developed method for tenatoprazole and its metabolites, as described elsewhere [7]. The analytical column was a Gemini C18 column (4.6 × 150 mm, 5 μm; Phenomenex), and the HPLC mobile phase consisted of acetonitrile and 5 mM potassium phosphate buffer (pH 7.3) (25:75, *v*/*v*). The substrate and its metabolites were detected at A_302_. The HPLC profile of the authentic compounds (i.e., tenatoprazole and its metabolites) were measured to compare to identify the major metabolites of tenatoprazole and tenatoprazole sulfide produced by HLMs [7].

The time course of the oxidation of tenatoprazole and tenatoprazole sulfide by HLMs (0.5 mg protein/mL) was analyzed to the estimated yield of metabolites and percentage of each metabolite using 0.60 mM substrate in 0.25 mL of 0.10 M potassium phosphate buffer (pH 7.4). After NGS was added to initiate the reaction, the samples were incubated for the designated time (i.e., 2, 5, 10, and 30 min, and 1, 2, 4, 8, 12, and 24 h) at 37 °C. Product formation was quantified using HPLC analysis, as mentioned above. 

Calibration standard curves of tenatoprazole were made using a blank control sample (a reaction mixture without substrate and IS) and 12 samples containing tenatoprazole and omeprazole (IS) covering the concentration ranges of 1–500 μM, as described previously [7]. The peak area ratio of tenatoprazole to IS was linear with respect to the analyte concentration over the range of 1–500 μM. The metabolites were quantified by comparing the peak areas of each metabolite to the mean peak areas of the IS (20 μM omeprazole). 

### 2.3. Hydroxylation of Tenatoprazole and Tenatoprazole Sulfide by Recombinant Human P450 Enzymes

The reaction mixture included 0.20 μM P450, 0.40 μM CPR, and tenatoprazole (0.10 mM) in 0.25 mL of 0.10 M potassium phosphate buffer (pH 7.4). The reaction was started by adding NGS and the incubation was continued for the indicated time at 37 °C. Incubation was then stopped by adding ice-chilled ethyl acetate (0.50 mL) and omeprazole (IS) was added to a final concentration of 20 μM. The mixtures were vortexed for 2 min, centrifuged at 1000× *g* for 20 min, and then the organic layer was taken for drying under nitrogen gas stream. The residues were dissolved in 180 μL of HPLC mobile phase and 30 μL was injected onto a Gemini C18 column (4.6 × 150 mm, 5 μm; Phenomenex, Torrance, CA, USA). The products formed were analyzed by HPLC at A_302_. The HPLC profile of the authentic chemicals (i.e., tenatoprazole, tenatoprazole sulfide, and their metabolites) were used to identify the major metabolites of tenatoprazole and tenatoprazole sulfide produced by the recombinant P450s [7].

The sulfoxidation reaction of tenatoprazole sulfide by CYP3A4 was measured by chiral chromatography [19]. The reaction mixture contained 0.20 μM CYP3A4, 0.40 μM CPR and tenatoprazole (0.10 mM) in 0.25 mL of potassium phosphate buffer (0.10 M, pH 7.4). The reaction was initiated by adding NGS at 37 °C. After 30 min of incubation, the reaction was stopped by adding cold ethyl acetate (0.50 mL) and dried under nitrogen gas stream. After the residues were solubilized in 180 μL mobile phase, 30 μL samples were injected onto a Chiralpak AD column (4.6 × 150 mm, 10 μm; Daicel Corporation, Osaka, Japan) with a hexane-ethanol mixture (30:70, *v*/*v*) as the mobile phase. The products formed were analyzed by HPLC at A_302_.

To analyze the steady-state kinetics (*k*_cat_ and *K*_M_) of each human P450 enzyme, the reaction mixture included 0.20 μM P450, 0.40 μM CPR, NGS, and tenatoprazole (1–500 μM). After the mixture was incubated at 37 °C for 5 min, the rate of product formation was determined using HPLC analysis, as mentioned above. The kinetic parameters (*k*_cat_ and *K*_m_) were determined by using a Michaelis–Menten nonlinear regression analysis with GraphPad Prism (GraphPad Software, San Diego, CA, USA).

### 2.4. Liquid Chromatography-Mass Spectrometry Analysis

To further characterize the nature of the metabolites, the reaction mixtures were additionally analyzed by a 6530 Quadrupole Time-of-Flight (QTOF) mass spectrometer (MS) (Agilent, Santa Clara, CA, USA) coupled with a 1290 Infinity ultra-performance liquid chromatography (UPLC) system (Agilent). The reaction mixture included 0.20 μM P450, 0.40 μM CPR, and 0.10 mM tenatoprazole in 0.25 mL potassium phosphate buffer (0.10 M, pH 7.4). The reaction was initiated by adding NGS at 37 °C. After 30 min of reaction, it was stopped by adding cold ethyl acetate (0.50 mL). Then, the upper layer was transferred to a glass tube and dried under nitrogen stream. The residues were reconstituted in 0.30 mL acetonitrile, then 1 μL samples were injected into an LC-MS system. The analytical column was a Hypersil Gold C18 column (100 × 2.1 mm, 1.9 μm; ThermoFisher Scientific). The HPLC mobile phase consisted of formic acid solution (0.1%, *v*/*v*) (A) and acetonitrile containing 0.1% formic acid (B). A gradient program was used for HPLC separation, and the flow rate was 0.4 mL/min. Mobile phase B had an initial composition of 10%, increased to 30% over 9 min, decreased to 10% over 0.2 min, and re-equilibrated to the initial condition for 0.8 min. Each run time was 10 min. The column and autosampler were kept at 35 °C and 4 °C, respectively, during analysis. Electrospray ionization was performed in the positive ion mode. The sheath gas flow rate was 8 L/min. The flow rate of drying gas was 12 L/min at 350 °C, and the nebulizer’s temperature was set at 350 °C. The capillary voltage was 4500 V in the positive mode, and that of the fragmentor was 175 V. Data acquisition were done over a scan range from 50 *m*/*z* to 1000 *m*/*z* in the centroid mode. A reference compound (C_18_H_18_O_6_N_3_P_3_F_24_; [M+H]^+^ = 922.0098) was used to correct all of the masses. Tandem mass spectrometry (MS/MS) data were acquired in profile mode via autoanalysis for structural characterization with 30 eV of the collision energy.

### 2.5. Identification of a New Major Metabolite of Tenatoprazole by NMR Spectroscopy

To characterize the unknown monohydroxylated metabolite, it was separated using HPLC, collected, and analyzed by nuclear magnetic resonance (NMR) spectroscopy. The reaction mixture contained 0.40 μM CYP3A4, 0.80 μM CPR, and tenatoprazole (1 mM) in 30 mL of potassium phosphate buffer (0.10 M, pH 7.4). An NGS was used to initiate the reaction. The reaction mixture was incubated for 2 h at 37 °C. The products formed were separated using analytical HPLC, as described above. The new metabolite, 1′-*N*-oxy-5′-hydroxytenatoprzole sulfide, had an isolated yield of 8.7%. NMR experiments were performed at 25 °C on an ProPulse 500 MHz NMR spectrometer (Agilent) equipped with a 5 mm AutoXDB probe. We used dimethyl sulfoxide-d_6_ as a solvent, and the chemical shifts for proton NMR spectra were obtained in parts per million (ppm) relative to tetramethylsilane. All of the NMR analysis were carried out with standard pulse sequences in the VNMRJ (v. 4.0) library and processed with the same software.

## 3. Results

### 3.1. Oxidation of Tenatoprazole and Tenatoprazole Sulfide by Human Liver Microsomes

Initially, to determine the ability of HLMs to oxidize tenatoprazole, the catalytic activity of HLMs toward tenatoprazole was assayed using a 0.10 mM substrate at 37 °C for 2 h. Figure 2A shows the HPLC profiles of tenatoprazole (peak 3) and its metabolites via HLMs. Five metabolites (peaks 1, 2, 4, 5, and 6) were observed. Among them, 5′-OH tenatoprazole (peak 1), 5′-OH tenatoprazole sulfide (peak 2), an uncharacterized monohydroxylated tenatoprazole (peak 4), and tenatoprazole sulfone (peak 5) were made in the presence of NADPH (i.e., NGS). Tenatoprazole sulfide (peak 6), the major metabolite, was formed spontaneously and nonenzymatically, even in the absence of NADPH, as observed previously [7]. Except tenatoprazole sulfone, four other metabolites were the same as the metabolites produced by bacterial CYP102A1 [7].

Because the major metabolite, tenatoprazole sulfide, could be oxidized further by P450 enzymes, the oxidation products of tenatoprazole sulfide from HLMs were examined by using them as a substrate (Figure 2B). Five metabolites (peaks 1–5) of tenatoprazole sulfide (peak 6) were observed: 5′-OH tenatoprazole (peak 1), 5′-OH tenatoprazole sulfide (peak 2), tenatoprazole (peak 3), a monohydroxylated tenatoprazole (peak 4), and tenatoprazole sulfone (peak 5) were made in the presence of NADPH. These results indicate that tenatoprazole and tenatoprazole sulfone are formed via sulfoxidation (*S*-oxygenation) reactions with tenatoprazole sulfide and tenatoprazole, respectively.

### 3.2. Analysis of Metabolites of Tenatoprazole by LC-MS/MS 

Ultra-performance liquid chromatography-electrospray ionization-QTOF-MS (UPLC-ESI-QTOF-MS) analysis was performed to obtain information about the metabolites of tenatoprazole and tenatoprazole sulfide. All of the identified metabolites of both tenatoprazole (Figure 3 and Table 1) and tenatoprazole sulfide (Appendix A) produced by HLMs are presented with their retention times, molecular formulars, protonated mass units, mass errors, and reaction types. The extracted ion chromatograms indicate that tenatoprazole was transformed into 5′-OH tenatoprazole, 5′-OH tenatoprazole sulfide, tenatoprazole sulfide, tenatoprazole sulfone, and two monohydroxylated products (Appendix A). Although sulfone was not detected in the HPLC trace, tenatoprazole sulfone was also detected in MS spectra.

We propose that the uncharacterized metabolite (peak 4 of Figure 2) is a dihydroxylated product at the nitrogen and methyl groups of the pyridine ring of tenatoprazole sulfide (Appendix A). However, the MS detected very low amounts of it (Figure 3 and Appendix A), although it showed apparent traces in HPLC (Figure 2). 

Furthermore, MS analyses of the metabolites of tenatoprazole catalyzed by CYP2C19 showed four metabolites formed by the hydroxylation of tenatoprazole and tenatoprazole sulfide (Appendix A). The uncharacterized hydroxylated metabolite (peak 4 of Figure 2) was a major product. For the CYP3A4-catalyzed tenatoprazole oxidation, the sulfone product was the major metabolite, in addition to the four minor metabolites and tenatoprazole sulfide (Appendix A). Here, very low amounts of the uncharacterized peak (M4) were also observed.

### 3.3. Characterization of a Monohydroxylated Metabolite of Tenatoprazole

In addition to three known metabolites (5′-OH tenatoprazole, 5′-OH tenatoprazole sulfide, and tenatoprazole sulfone), a mono-hydroxylated product (peak 4 of Figure 2) was found, which is a new major metabolite of the metabolic pathways of both tenatoprazole and tenatoprazole sulfide. The chemical structure of this new metabolite was identified to understand the metabolic pathways of both compounds. After the peak was collected from the eluent and dried, its chemical structure was determined by NMR analyses (Appendix A, Figure 4 and Appendix A).

Spectral assignments of the metabolite obtained from the tenatoprazole were carried out by ^1^H NMR (Figure 4A) and one-dimensional selective NOE experiments (Figure 4B, Appendix A) [7,20]. Chemical shifts of the metabolite (peak 4 of Figure 2) are listed in Appendix A. Important NOE results for identifying the monohydroxylated metabolite are shown in Figure 4B. The metabolite was identified as 1′-*N*-oxy-5′-hydroxytenatoprazole sulfide.

### 3.4. Time-Course Profiles of Tenatoprazole and Tenatoprazole Sulfide Oxidation by Human Liver Microsomes

To study the metabolic profiles during oxidation of tenatoprazole and tenatoprazole sulfide by HLMs, time course measurements of each compound were performed at 500 μM for 24 h (Figure 5). The concentration of tenatoprazole as a substrate progressively decreased during the reaction time; only 1.0% of tenatoprazole remained after 12 h of incubation (Figure 5A, right panel). Simultaneously, the concentrations of the major products tenatoprazole sulfide, 1′-*N*-oxy-5′-OH tenatoprazole sulfide, 5′-OH tenatoprazole sulfide, and tenatoprazole sulfone increased to 62, 21, 10, and 4.2%, respectively, after 12 h, and the concentration of 5′-OH tenatoprazole increased slightly (0.96% after 12 h of incubation). When tenatoprazole was incubated with the reaction mixtures without NADPH (Figure 5A, left panel), the tenatoprazole concentration rapidly decreased over the incubation time. After 12 h, only 4.4% of the tenatoprazole remained. However, the conversion yield to tenatoprazole sulfide was very high, at 94.8%. These results indicate that much of the tenatoprazole spontaneously degraded to tenatoprazole sulfide without any other metabolites forming. 

Furthermore, the results suggest that tenatoprazole sulfide can act as a substrate for P450-mediated reactions. Another possibility is that 5′-OH tenatoprazole sulfide can be derived from 5′-OH tenatoprazole in a non-enzymatic manner. When tenatoprazole sulfide was oxidized by HLMs in the presence of NADPH, 1′-*N*-oxy-5′-OH tenatoprazole sulfide, 5′-OH tenatoprazole sulfide, and tenatoprazole sulfone increased to 9.3, 8.2, and 2.1%, respectively, after 12 h of incubation (Figure 5B, right panel). Only 0.49% and 0.48% of the tenatoprazole and 5′-OH tenatoprazole remained after the incubation, respectively. After 12 h of reaction, 79.4% of the tenatoprazole sulfide remained. More than 99% of the tenatoprazole sulfide remained after 12 h of incubation in the absence of NADPH (Figure 5B, left panel). This indicates that tenatoprazole sulfide is very stable, in contrast to tenatoprazole. The reactions of tenatoprazole and tenatoprazole sulfide appeared to be reliable for up to 8 h when their reaction rates reached their maximum.

### 3.5. Tenatoprazole and Tenatoprazole Sulfide Oxidation by Recombinant Human P450 Enzymes

CYP2C19 and CYP3A4 catalyze the formation of 5′-OH tenatoprazole and tenatoprazole sulfone, respectively [5]. Table 2 shows the steady-state kinetics of the metabolites’ formation catalyzed by the two recombinants of CYP3A4 and CYP2C19. CYP2C19 had *k*_cat_ values of 4.0 and 2.0 min^−1^ and *K*_m_ values of 23 and 70 μM for the formation of 5′-OH tenatoprazole and 5′-OH tenatoprazole sulfide, respectively. Although CYP3A4 had a higher *k*_cat_ value (6.3 min^−1^) than CYP2C19 did for 5′-OH tenatoprazole formation, it showed a 100-fold higher *K*_m_ value (2.3 mM) than CYP2C19 did. Therefore, the CYP2C19-catalyzed efficiency for 5′-OH tenatoprazole formation was 63-fold higher than that of CYP3A4. This result suggests that CYP2C19 is the major contributor to 5′-OH tenatoprazole production in human liver. 

CYP2C19 seems to be the major P450 enzyme used to make 5′-OH tenatoprazole sulfide, and two possible pathways exist to make it: 5′-OH tenatoprazole can be non-enzymatically or spontaneously converted to 5′-OH tenatoprazole sulfide, and/or non-enzymatically formed tenatoprazole sulfide can be hydroxylated to 5′-OH tenatoprazole sulfide. Tenatoprazole sulfone formation was mediated by CYP3A4, with a *k*_cat_ value of 2.7 min^−1^ and a *K*_m_ value of 4.0 mM. However, CYP2C19 did not catalyze the sulfoxidation of tenatoprazole. 

When compared to omeprazole oxidation by human P450s [21], both CYPs 2C19 and 3A4 show similar oxidation patterns. Both enzymes involve 5′-hydroxylation reaction, although only CYP3A4 showed sulfoxidation activity. Although 1′-*N*-oxy-5′-hydroxytenatoprazole sulfide was the major metabolite from longer incubation periods in HLMs (Figure 5) and recombinant CYP2C19 (Appendix A), only trace amounts of this metabolite were observed in CYP3A4-catalyzed reaction under the experimental conditions (30 min incubation) (Appendix A). On the other hand, CYPs 1A2, 1B1, 2A6, and 2E1 did not show the apparent formation (<0.05 nmol product/min/nmol P450) of major metabolites (i.e., 5′-OH tenatoprazole, 5′-OH tenatoprazole sulfide, and tenatoprazole sulfone) (Appendix A). However, CYP2C9 and CYP2D6 showed very low formation rates of 5′-OH tenatoprazole (both 0.1 min^−1^) and 5′-OH tenatoprazole sulfide (0.1 and 0.3 min^−1^, respectively). At present, the possible contribution of CYP2C9 and CYP2D6 to the metabolism of tenatoprazole cannot be completely excluded.

During the oxidation reactions for tenatoprazole with CYP2C19 and CYP3A4 (Appendix A), we found that only CYP3A4 could catalyze the sulfoxidation reaction. When tenatoprazole sulfide was used as a substrate for the sulfoxidation, CYP3A4 but not CYP2C19 showed the sulfoxidation activity required to produce tenatoprazole. To examine the preference of chiral products, chiral column chromatography was performed with the tenatoprazole sulfide and CYP3A4 (Figure 6 and Table 3). The formation of *S*- and *R*-tenatoprazole had *k*_cat_ values of 2.7 and 4.0 min^−1^ and *K*_m_ values of 66 and 59 μM, respectively. Therefore, the catalytic efficiency of *R*-tenatoprazole formation was 1.7-fold higher than that of *S*-tenatoprazole.

Chiral chromatography was performed on the tenatoprazole oxidation samples from the HLMs to examine the presence of chiral metabolites (Appendix A). We collected, concentrated, and analyzed the peaks (peaks 1–4 and 6 of Figure 2A) found in chiral chromatography (Appendix A). Samples from peaks 1 and 3 showed HPLC traces, corresponding to 5′-OH tenatoprazole and tenatoprazole, respectively. Peak 4, 1′-*N*-Oxy-5′-OH tenatoprazole sulfide, did not show any peaks, as expected.

## 4. Discussion

In this work, we studied how human P450s affect the metabolic fate of tenatoprazole and its sulfide, a major metabolite of tenatoprazole, using HLMs and human P450 enzymes. Recombinant CYP 2C19 and 3A4 showed distinct regioselective monooxygenase activities toward tenatoprazole and its sulfide to produce 5′-OH tenatoprazole, 5′-OH tenatoprazole sulfide, and tenatoprazole sulfone. In particular, we identified a new major metabolite of tenatoprazole sulfide, 1′-*N*-oxy-5′-hydroxytenatoprzole sulfide, which had never been reported on before. Although many PPI drugs share the pyridinyl structure, *N*-oxide metabolites of the pyridyl group have never been reported on before. 

First of all, we should consider the spontaneous desulfoxidation of tenatoprazole to produce tenatoprazole sulfide, even at neutral pH. After incubating tenatoprazole for 6 h without oxidation by HLMs, 87% of the tenatoprazole was spontaneously converted to its sulfide form (Figure 5). When tenatoprazole with HLMs was incubated for 6 h in the presence of NADPH, 60% of the tenatoprazole was converted to tenatoprazole sulfide, which indicates some portion of tenatoprazole sulfide was converted to other products by P450-catalyzed reactions (Figure 5). This indicates that the sulfide form can be used as a substrate when tenatoprazole is administered. A futile cycle of tenatoprazole and tenatoprazole sulfide consuming NADPH by CYP3A4-catalyzed reaction is also possible: a spontaneous desulfoxidation of tenatoprazole to form tenatoprazole sulfide and, in turn, a sulfoxidation reaction of tenatoprazole sulfide catalyzed by CYP3A4 produces tenatoprazole. When tenatoprazole is administered, tenatoprazole sulfide should be considered another substrate for P450-catalyzed reaction. An in vivo racemization reaction is also possible when a chiral enantiomer is administered (Figure 6).

Because tenatoprazole sulfide should be a major metabolite (>10%), its metabolic fate should be studied during the drug-development process, based on the FDA’s strategy [8,9]. Its efficacy as a PPI and its toxicity should be studied during the drug development process.

Taking everything into consideration, the spontaneous or non-enzymatic conversion of tenatoprazole to tenatoprazole sulfide at a neutral pH should be considered for pharmacokinetics of tenatoprazole metabolism. Specifically, 5′-OH tenatoprazole sulfide can be made via the direct hydroxylation of tenatoprazole sulfide or nonenzymatic desulfoxidation of 5′-OH tenatoprazole. Because of the nonenzymatic conversion of sulfoxide to the sulfide group of tenatoprazole at neutral pH, the metabolic fate of tenatoprazole is more complicated than that of omeprazole, in that omeprazole is not spontaneously or nonenzymatically degraded to omeprazole sulfide at neutral pH [22]. The interconversion of sulfoxide and sulfide groups through P450-catalyzed and nonenzymatic reactions makes it difficult to understand how tenatoprazole is metabolized in the human body. Taken together, a possible scheme for tenatoprazole metabolism in HLMs can be proposed. Tenatoprazole and tenatoprazole sulfide can be used as substrates with eight end products, based on this study’s results (Figure 7).

Most of the human metabolites of tenatoprazole we compared were very similar to CYP102A1 metabolites [7]. This indicates the relevance of bacterial CYP102A1 as a prototype P450 enzyme for studying human P450 enzymes [23,24,25,26].

The two enantiomers of tenatoprazole, *R* and *S*, show notably different properties of pharmacokinetics. Therefore, it is possible to prepare medicinal products having specific activities by isolating each enantiomer, which have different pharmacokinetic profiles from that of the previously known racemic mixture. Use of each enantiomer would make it possible do the precise indications for treating perfectly identified pathologies. The racemic mixture and the *S*-isomer of tenatoprazole differ meaningfully in their pharmacokinetic properties [5]. These differences are essential because they would provide clinicians with medicinal products for the effective treatment of determined pathologies. More specifically, the enantiomers have different pharmacokinetic parameters and elimination half-lives. At the same time, a chiral pharmacokinetic study showed the superiority of the *S*-isomer in the racemic tenatoprazole in terms of pharmacokinetic characteristics. 

A main issue with PPI metabolism is that PPIs are mostly metabolized by CYP2C19, which exhibits genetic polymorphism. The *R*-isomer is metabolized predominantly by CYP2C19, whereas the *S*-isomer is metabolized by both CYP2C19 and CYP3A4. In this regard, CYP3A4 can compensate for any deficiency or blockade of CYP2C19. Indeed, the *S*-isomer in a slow metabolizer will be metabolized by CYP3A4, leading to uniform pharmacokinetic parameters regardless of whether the patient is a slow or rapid metabolizer. These observations have led to the proposal of isolating and administering just the *S*-isomer, which would have several benefits over administering the racemic mixture and *R-*isomer when treating digestive pathologies, which needs strong and prolonged inhibition of acid secretion.

Furthermore, tenatoprazole and other PPIs are being developed as novel drug candidates in new therapeutic areas as anti-virus [27,28] and anti-cancer treatments [29,30]. Studying the metabolites of tenatoprazole is also very important for this new approach.

## 5. Conclusions

Tenatoprazole, a newly developed PPI, was developed as an acid inhibitor for treating gastric acid hypersecretion disorders such as gastric ulcer and reflux esophagitis. Tenatoprazole is oxidized into three major metabolites in human livers—5′-OH tenatoprazole, tenatoprazole sulfone, and tenatoprazole sulfide. While CYP2C19 prefers the C-5′ hydroxylation of tenatoprazole, CYP3A4 primarily catalyzes the sulfoxidation reaction to make tenatoprazole sulfone. Tenatoprazole sulfide is formed spontaneously and non-enzymatically. However, its metabolic fate in the human liver has not been fully characterized. In the present study, the roles of human P450 enzymes in the metabolism of tenatoprazole and tenatoprazole sulfide were characterized using HLMs and recombinant human P450s. CYPs 2C19 and 3A4 showed distinct regio-specific and stereo-specific monooxygenase activities toward tenatoprazole and tenatoprazole sulfide, respectively. A new major metabolite of tenatoprazole sulfide was characterized: 1′-*N*-oxy-5′-hydroxytenatoprzole sulfide, which has not been reported on before. In conclusion, the metabolic fates of tenatoprazole and tenatoprazole sulfide should be considered in the clinical use of tenatoprazole. Furthermore, possible drug-drug interactions should be considered because CYP2C19 typically exhibits genetic polymorphism, and inter-individual differences in CYP3A4 are well known. 

## Figures and Tables

**Figure 1 pharmaceutics-15-00023-f001:**
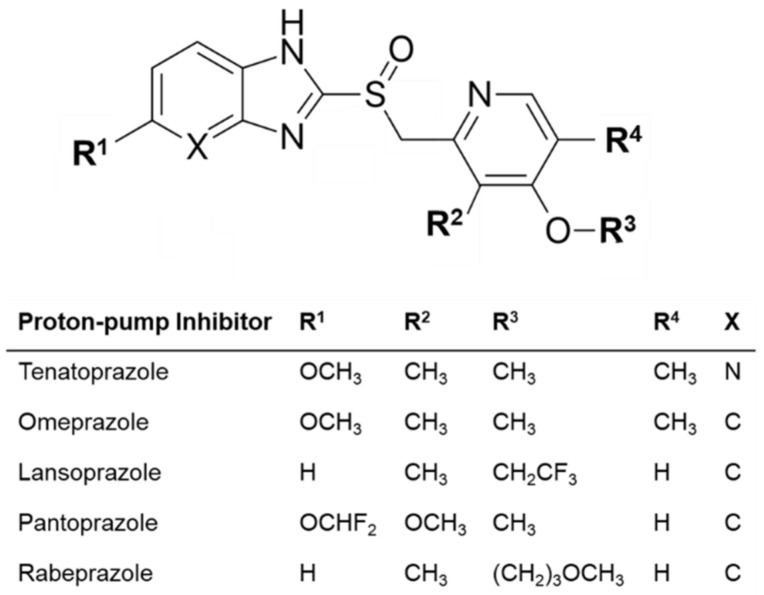
Chemical structures of common proton pump inhibitors.

**Figure 2 pharmaceutics-15-00023-f002:**
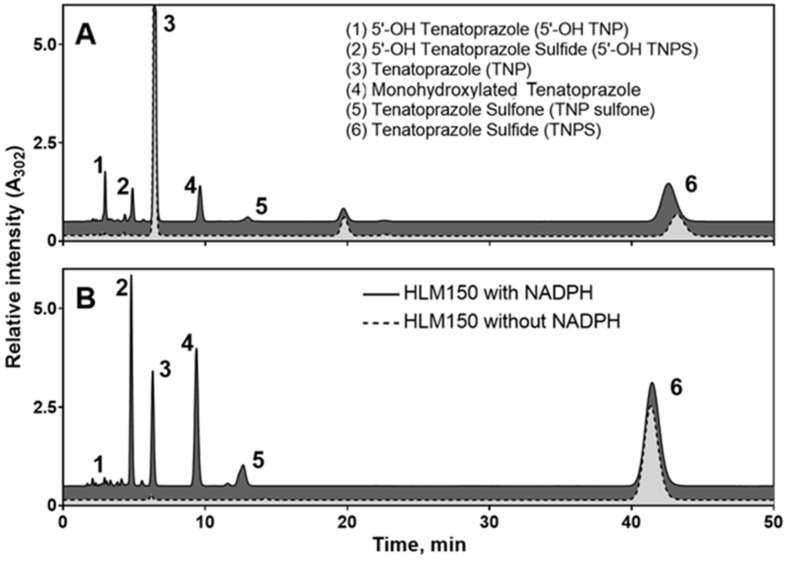
HPLC chromatograms of the metabolites of tenatoprazole (**A**) and tenatoprazole sulfide (**B**) produced by HLMs after a two-hour incubation. Solid and dotted lines indicate the presence and absence of NADPH, respectively.

**Figure 3 pharmaceutics-15-00023-f003:**
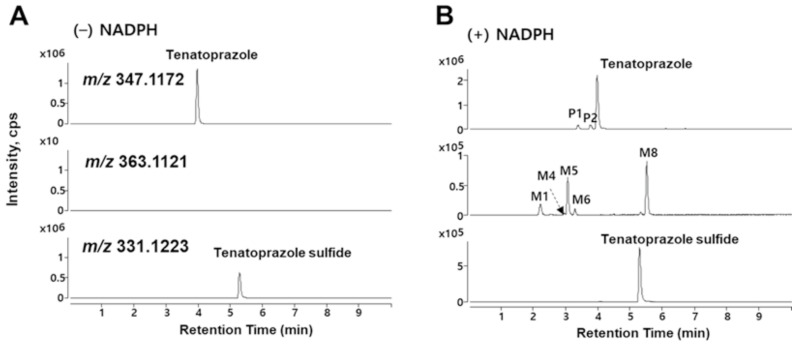
LC-QTOF-MS analyses of the metabolites from the tenatoprazole oxidized by HLMs. The extracted ion chromatograms were constructed by incubating tenatoprazole with HLMs in the absence (**A**) and presence (**B**) of NADPH.

**Figure 4 pharmaceutics-15-00023-f004:**
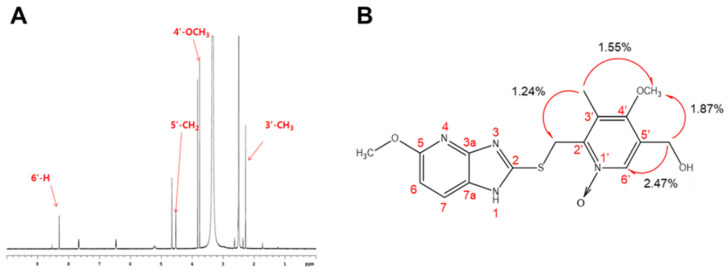
Characterization of a major metabolite of tenatoprazole, 1′-*N*-oxy-5′-hydroxytenatoprazole sulfide. (**A**) ^1^H NMR spectrum of 1′-*N*-oxy-5′-hydroxytenatoprazole sulfide corresponding to peak 4 of Figure 2 and metabolite M4 of Table 1 and Appendix A. (**B**) Key NOE results of 1′-*N*-oxy-5′-hydroxytenatoprazole sulfide.

**Figure 5 pharmaceutics-15-00023-f005:**
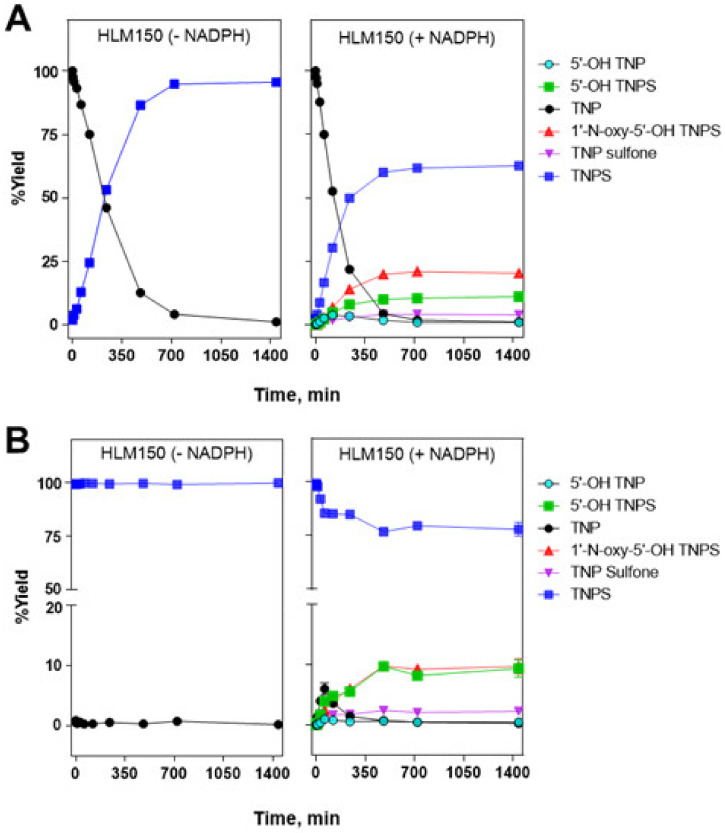
Time course profiles of oxidation of tenatoprazole (**A**) and tenatoprazole sulfide (**B**) by HLMs. Each substrate (500 μM) was incubated with HLMs for the indicated times in the absence of NADPH (**left** panels) and the presence of NADPH (**right** panels). The values are shown as the means ± SD of triplicate determination. Tenatoprazole and tenatoprazole sulfide are labeled as TNP and TNPS, respectively, as shown at Figure 2A.

**Figure 6 pharmaceutics-15-00023-f006:**
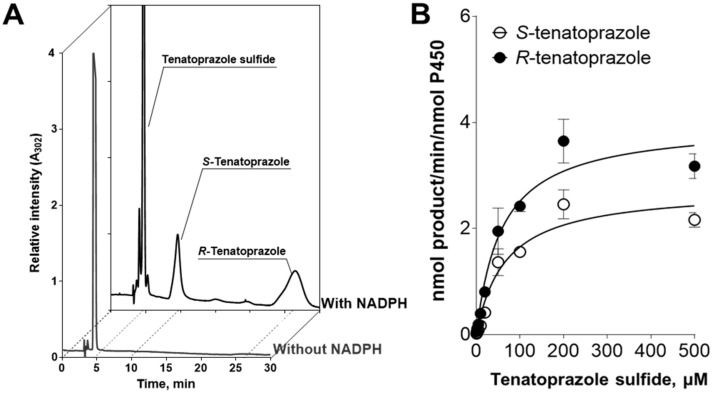
Sulfoxidation of tenatoprazole sulfide by CYP3A4. (**A**) HPLC chromatogram of the products of tenatoprazole sulfide oxidation by CYP3A4. (**B**) Effect of substrate concentration on tenatoprazole sulfide sulfoxidation by CYP3A4.

**Figure 7 pharmaceutics-15-00023-f007:**
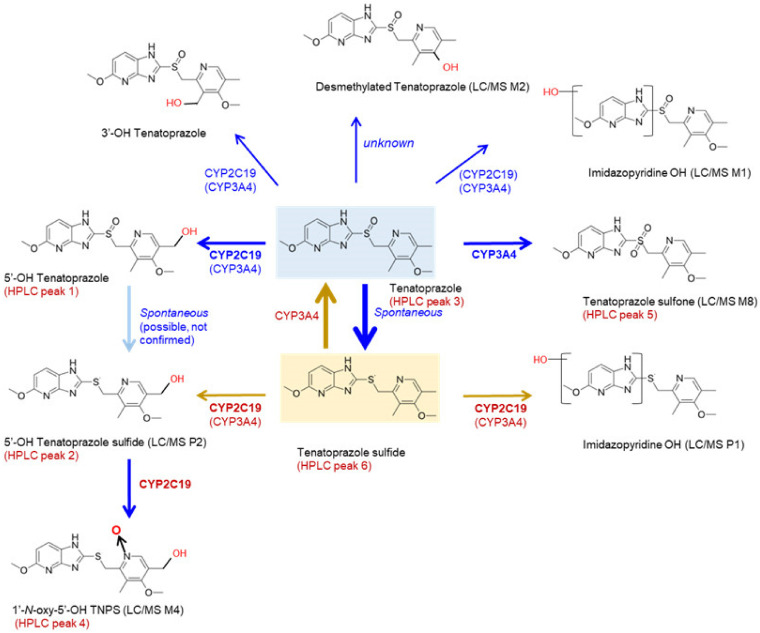
A possible scheme for tenatoprazole metabolism in human liver microsomes. Notes were added to match the HPLC metabolites (Figure 2) with LC-QTOF metabolite (Appendix A). Nonenzymatic spontaneous desulfoxidation of 5′-OH tenatoprazole to 5′-OH tenatoprazole sulfide is possible but experimental evidence was not provided in this study.

**Table 1 pharmaceutics-15-00023-t001:** Characteristics of tenatoprazole metabolites of HLMs by LC-QTOF-MS.

Metabolites	Reaction Type	Formula	Theoretical Mass [M + H]^+^	Measured Mass [M + H]^+^	Fragmentation (*m*/*z*)	Error (δ) ppm	RT (min)	Abs. Intensity
M1	Hydroxylation	C_16_H_18_N_4_O_4_S	363.1121	363.115	330, 198, 180, 151, 136	8.2	2.2	87,257
M4	Desulfoxidation Hydroxylation *N*-Oxygenation	C_16_H_18_N_4_O_4_S	363.1121	363.115	345, 312, 214, 196, 152	9.2	3.0	9752
M5	Hydroxylation	C_16_H_18_N_4_O_4_S	363.1121	363.115	214, 196, 184, 167, 152	10.3	3.1	235,348
M6	Hydroxylation	C_16_H_18_N_4_O_4_S	363.1121	363.116	214, 196, 184, 167, 152	11.2	3.3	37,006
P1	Desulfoxidation Hydroxylation	C_16_H_18_N_4_O_3_S	347.1172	347.122	314, 182, 150, 136, 120	13.0	3.4	614,468
P2	Desulfoxidation Hydroxylation	C_16_H_18_N_4_O_3_S	347.1172	347.121	314, 198, 150, 138, 121	12.3	3.8	689,875
Tenatoprazole sulfide	Desulfoxidation	C_16_H_18_N_4_O_2_S	331.1223	331.124	298, 283, 182, 150, 136	6.6	5.3	3,607,768
M8	*S*-Oxygenation	C_16_H_18_N_4_O_4_S	363.1121	363.115	299, 214, 196, 150	9.1	5.5	347,615
Tenatoprazole	Parent	C_16_H_18_N_4_O_3_S	347.1172	347.121	198, 180, 168, 151, 136	12.9	4.0	4,332,256

**Table 2 pharmaceutics-15-00023-t002:** Kinetic parameters of tenatoprazole oxidation reactions by human P450s.

Human P450	Product	*k*_cat_ (min^−1^)	*K*_m_ (µM)	*k*_cat_/*K*_m_ (min^−1^·µM^−1^)
2C19	5′-OH tenatoprazole	4.0 ± 0.2	23 ± 3	0.17 ± 0.02
2C19	5′-OH tenatoprazole sulfide	2.0 ± 0.1	70 ± 5	0.029 ± 0.003
3A4	5′-OH tenatoprazole	6.3 ± 0.3	2300 ± 170	0.0027 ± 0.0002
3A4	Tenatoprazole sulfone	2.7 ± 0.8	4000 ± 1600	0.00068 ± 0.00035

**Table 3 pharmaceutics-15-00023-t003:** Kinetic parameters of tenatoprazole sulfide oxidation by human CYP3A4.

Product	*k*_cat_ (min^−1^)	*K*_m_ (µM)	*k*_cat_ /*K*_m_ (min^−1^·µM^−1^)
*S*-tenatoprazole	2.7 ± 0.2	66 ± 16	0.041 ± 0.009
*R*-tenatoprazole	4.0 ± 0.3	59 ± 13	0.068 ± 0.012

## Data Availability

Data is contained within the article or Appendix A.

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
