# Peer review of "Roles of Human Liver Cytochrome P450 Enzymes in Tenatoprazole Metabolism"

_pharmaceutics, 2022, doi:10.3390/pharmaceutics15010023_

Round 1
Reviewer 1 Report
The author investigated the roles of human P450 enzymes in the metabolic pathway of tenatoprazole and tenatoprazole sulfide using human liver microsomes and recombinant human P450 enzymes. The results showed that CYP2C19 and CYP3A4 show distinct regioselective and stereospecific monooxygenase activities toward tenatoprazole and tenatoprazole sulfide, respectively. Furthermore, a new major metabolite of tenatoprazole sulfide was found—1’-N-oxy-5’ hydroxytenatoprzole sulfide, which has never been reported. In conclusion, the metabolic fates of tenatoprazole and tenatoprazole sulfide should be considered in the clinical use of tenatoprazole. On the whole, the work is well organized, and the conclusion is supported by the data in this work. But there are still some problems with this manuscript.
Major comments and revision suggestions as follows,
1. In general, human liver microsomal (HLM) will be inactivated after 2 hours. In this paper, whether the reaction for 12 hours is reliable?
2. We observe that in figure 2, in addition to M1-M6, there is also a peak at 20 min in figure A. what is it?
3. It can be observed from figure 6, the efficiency of R-tenatoprazole formation was 1.7-fold higher than that of S-tenatoprazole. Please pay attention to the accuracy of the statement in this article.
4. Mentioned in ‘3.5’ ‘CYPs 1A2, 1B1, 2E1, 2D6, and 2A6 did not show the apparent formation of any major metabolites’. Is there any data support?
5. In figure 7, 5’-OH tenatoprazole sulfide can be made via the direct hydroxylation of tenatoprazole sulfide or nonenzymatic desulfoxidation of 5’-OH tenatoprazole. Is there any evidence to support this part?
6. There should be a space between the value and the unit. Please check and modify it.
7. The table comments are centered and the format should be consistent. Please pay attention to the modification.
8. In 'Km',' m'is italic, lowercase letters. Please pay attention to writing mistakes.
Author Response
Comments and Suggestions for Authors
The author investigated the roles of human P450 enzymes in the metabolic pathway of tenatoprazole and tenatoprazole sulfide using human liver microsomes and recombinant human P450 enzymes. The results showed that CYP2C19 and CYP3A4 show distinct regioselective and stereospecific monooxygenase activities toward tenatoprazole and tenatoprazole sulfide, respectively. Furthermore, a new major metabolite of tenatoprazole sulfide was found—1’-N-oxy-5’ hydroxytenatoprzole sulfide, which has never been reported. In conclusion, the metabolic fates of tenatoprazole and tenatoprazole sulfide should be considered in the clinical use of tenatoprazole. On the whole, the work is well organized, and the conclusion is supported by the data in this work. But there are still some problems with this manuscript.
Major comments and revision suggestions as follows,
- In general, human liver microsomal (HLM) will be inactivated after 2 hours. In this paper, whether the reaction for 12 hours is reliable?
(Response) According to the time course profiles of oxidation of tenatoprazole (A) and tenatoprazole sulfide (B) by HLMs as shown at Figure 5, the reaction seems to be reliable up to 8 hours. It was mentioned in the revised manuscript.
[Page 8, 2nd para, Line 11-13 of revised manuscript] This now reads “By the way, the reactions of tenatoprazole and tenatoprazole sulfide appeared to be reliable for up to 8 h when their reaction rates reached their maximum.”
- We observe that in figure 2, in addition to M1-M6, there is also a peak at 20min in figure A. what is it?
(Response) The peak that appeared at 20 min in Figure 2A was also observed in the control sample incubated without NADPH, suggesting that this peak was not a metabolite generated by P450s. Therefore, we did not characterize the structure of this peak.
- It can be observed from figure 6, the efficiency of R-tenatoprazole formation was 1.7-fold higher than that of S-tenatoprazole. Please pay attention to the accuracy of the statement in this article.
(Response) We agree with the reviewer’s criticism. It was a typo.
[Page 10, 2nd para, Line 8-9 of revised manuscript] This now reads “Therefore, the catalytic efficiency of R-tenatoprazole formation was 1.7-fold higher than that of S-tenatoprazole.”
- Mentioned in ‘3.5’ ‘CYPs 1A2, 1B1, 2E1, 2D6, and 2A6 did not show the apparent formation of any major metabolites’. Is there any data support?
(Response) Accordingly, the data of CYPs 1A2, 1B1, 2A6, 2C9, 2D6, and 2E1 were shown at Figure S6. CYPs 1A2, 1B1, 2A6, and 2E1 did not show the apparent formation (<0.05 nmol product/min/nmol P450) of major metabolites (i.e., 5’-OH tenatoprazole, 5’-OH tenatoprazole sulfide, and tenatoprazole sulfone). However, CYPs 2C9 and 2D6 showed very low catalytic activity (0.1 ~ 0.3 min-1) with 60 min incubation. The result has been described in the revised manuscript.
[Page 10, Line 10-13 of revised manuscript] This now reads ” On the other hand, CYPs 1A2, 1B1, 2A6, and 2E1 did not show the apparent formation (<0.05 nmol product/min/nmol P450) of major metabolites (i.e., 5’-OH tenatoprazole, 5’-OH tenatoprazole sulfide, and tenatoprazole sulfone) (Figure S6). However, CYP2C9 and CYP2D6 showed very low formation rates of 5’-OH tenatoprazole (both 0.1 min-1) and 5’-OH tenatoprazole sulfide (0.1 and 0.3 min-1, respectively). At present, a possibility of CYP2C9 and CYP2D6 contributed to the metabolism of tenatoprazole cannot be completely excluded.”
- In figure 7, 5’-OH tenatoprazole sulfide can be made via the direct hydroxylation of tenatoprazole sulfide or nonenzymatic desulfoxidation of 5’-OH tenatoprazole. Is there any evidence to support this part?
(Response) We agree with the reviewer’s criticism. 5’-OH tenatoprazole sulfide can be made via the direct hydroxylation of tenatoprazole sulfide, as shown in Figures 2B, 5B, Table S2, Figure S1, and Figure S3. However, we did not provide evidence to support the nonenzymatic desulfoxidation of 5’-OH tenatoprazole in this study. We discussed the possibility of nonenzymatic spontaneous desulfoxidation of 5’-OH tenatoprazole.
[Page 8, 2nd para, Line 2-3 of revised manuscript] “Another possibility is that 5’-OH tenatoprazole sulfide is non-enzymatically formed from 5’-OH tenatoprazole.”
[Page 9, 2nd para, Line 1-4 of revised manuscript] “CYP2C19 seems to be the major P450 enzyme used to make 5’-OH tenatoprazole sulfide, and two possible pathways exist to make it: 5’-OH tenatoprazole can be non-enzymatically converted to 5’-OH tenatoprazole sulfide, and/or non-enzymatically formed tenatoprazole sulfide can be hydroxylated to 5’-OH tenatoprazole sulfide.”
[P12, legend of Figure 7 in revised manuscript] “Nonenzymatic spontaneous desulfoxidation of 5’-OH tenatoprazole to 5’-OH tenatoprazole sulfide is possible but experimental evidence was not provided in this study.”
- There should be a space between the value and the unit. Please check and modify it.
(Response) We appreciate the reviewer’s comments. We checked the entire manuscript to correct them.
- The table comments are centered and the format should be consistent. Please pay attention to the modification.
(Response) We appreciate the reviewer’s comments. We checked the entire manuscript to correct them.
- In 'Km',' m'is italic, lowercase letters. Please pay attention to writing mistakes.
(Response) We appreciate the reviewer’s comments regarding Km. We have corrected the revised manuscript.
Reviewer 2 Report
In present paper, cytochrom p450 metabolism of tenatoprazole , a proton pump inhibitor, which has prolonged duration of action was evaluated. New major metabolites, 1'-N-Oxy-5'hydroxy tenatoprazol sulfide was identified using hepatic microsomal/recombinant p450 incubation and LC-QTOF detection. Overall structure of the study was concrete and well prepared. There is some concerns and minor corrections required in below.
1. some information of metabolite indication is confusing. Could authors matching HPLC metabolite with LC-QTOF metabolite (as the figure 7).
2. How about contribution of other major cyp isoforms which has impotrant role in xenobiotic metabolism (CYP1A2, 2B6, 2C8, 2C9). Especially, CYP2C8 or 2C9 could have overapped seletivity with 2C19.
3. in Figure 5, same abbreviation used (TNPS), it is very hard to understand what is the correct metabolite. please make same abbreviation of metabolites in the paper.
4. Is the P450 metabolism major contribution of tenaprazole metabolism? how about other phase 2 conjugation such as UGT glucuronidation ? In addition, is there any control experiment of tenatporzole to tenatoprazole sulfide formation without HLM ? is it possible other NADPH indepedent enzymatic reaction?
minor points,
P4 sample amount looks typo. It should be correct, μL?
P7 Figure 3, with QTOF (HRMS) m/z should be presented with with 4 decimal points (or correct width of the extractied m/z of the chromatogram).
P9 Figure, correct abbreviation. Recommedation of color graphs than mono-color
P11 please replace 'efficacy-as' by 'efficacy as'
Author Response
Reviewer #2
Comments and Suggestions for Authors
In present paper, cytochrom p450 metabolism of tenatoprazole , a proton pump inhibitor, which has prolonged duration of action was evaluated. New major metabolites, 1'-N-Oxy-5'hydroxy tenatoprazol sulfide was identified using hepatic microsomal/recombinant p450 incubation and LC-QTOF detection. Overall structure of the study was concrete and well prepared. There is some concerns and minor corrections required in below.
- some information of metabolite indication is confusing. Could authors matching HPLC metabolite with LC-QTOF metabolite (as the figure 7).
(Response) We appreciate the reviewer’s valuable comments. We added notes to match the HPLC metabolites with LC-QTOF metabolites in Figure 7, as commented. Accordingly, revised Figure 7 was added to the revised manuscript.
- How about contribution of other major cyp isoforms which has impotrant role in xenobiotic metabolism (CYP1A2, 2B6, 2C8, 2C9). Especially, CYP2C8 or 2C9 could have overapped seletivity with 2C19.
(Response) We agree with the reviewer’s criticism regarding other major CYPs in the human liver. Accordingly, the data of CYPs 1A2, 1B1, 2A6, 2C9, 2D6, and 2E1 were shown at Figure S6. CYPs 1A2, 1B1, 2A6, and 2E1 did not show the apparent formation (<0.05 nmol product/min/nmol P450) of major metabolites (i.e., 5’-OH tenatoprazole, 5’-OH tenatoprazole sulfide, and tenatoprazole sulfone). However, CYPs 2C9 and 2D6 showed very low catalytic activity (0.1 ~ 0.3 min-1) with 60 min incubation. The result has been described.
[Page 10, Line 7-13 of revised manuscript] This now reads ”On the other hand, CYPs 1A2, 1B1, 2A6, and 2E1 did not show the apparent formation (<0.05 nmol product/min/nmol P450) of major metabolites (i.e., 5’-OH tenatoprazole, 5’-OH tenatoprazole sulfide, and tenatoprazole sulfone) (Figure S6). However, CYP2C9 and CYP2D6 showed very low formation rates of 5’-OH tenatoprazole (both 0.1 min-1) and 5’-OH tenatoprazole sulfide (0.1 and 0.3 min-1, respectively). At present, a possible contribution of CYP2C9 and CYP2D6 to the metabolism of tenatoprazole cannot be completely excluded.”
- in Figure 5, same abbreviation used (TNPS), it is very hard to understand what is the correct metabolite. please make same abbreviation of metabolites in the paper.
(Response) “TNPS (third line)” in Figure 5A was a typo. It has been corrected to “TNP” as commented. Accordingly, Figure 5 has been redrawn to avoid any confusion.
We used the abbreviations for tenatoprazole (TNP) and tenatoprazole sulfide (TNPS) in the figures because the names are too long at figures. However, we want to use the full spelling for them because the abbreviations are not generally used.
- Is the P450 metabolism major contribution of tenaprazole metabolism? how about other phase 2 conjugation such as UGT glucuronidation ? In addition, is there any control experiment of tenatporzole to tenatoprazole sulfide formation without HLM ? is it possible other NADPH indepedent enzymatic reaction?
(Response) At present, P450-catalyzed metabolism is known as the major contributor to tenatoprazole metabolism. To our best knowledge, there are no reports regarding the Phase II metabolism of tenatoprazole up to now.
We previously reported the control experiment of tenatporzole to tenatoprazole sulfide formation without HLM [ref. #6 in revised manuscript]. Please refer Figure S8
The spontaneous formation of sulfide metabolites from sulfoxide compounds such as tenatoprazole and 5’-OH tenatoprazole seems to be possible without NADPH. However, any other NADPH-independent enzymatic reactions have not been reported.
minor points,
P4 sample amount looks typo. It should be correct, μL?
(Response) The typo has been corrected as commented.
P7 Figure 3, with QTOF (HRMS) m/z should be presented with with 4 decimal points (or correct width of the extractied m/z of the chromatogram).
(Response) m/z values have been denoted with 4 decimal points as indicated.
P9 Figure, correct abbreviation. Recommedation of color graphs than mono-color
(Response) “TNPS (third line)” in Figure 5A was a typo. It has been corrected to “TNP” as commented.
[Figure 5 in revised manuscript] Figure 5 has been redrawn as color graphs, as commented.
P11 please replace 'efficacy-as' by 'efficacy as'
(Response) It has been corrected as commented.